# Subacromial Bursa: A Neglected Tissue Is Gaining More and More Attention in Clinical and Experimental Research

**DOI:** 10.3390/cells11040663

**Published:** 2022-02-14

**Authors:** Franka Klatte-Schulz, Kathi Thiele, Markus Scheibel, Georg N. Duda, Britt Wildemann

**Affiliations:** 1Julius Wolff Institute, Berlin Institute of Health at Charité-Universitaetsmedizin Berlin, 13353 Berlin, Germany; georg.duda@bih-charite.de (G.N.D.); britt.wildemann@bih-charite.de (B.W.); 2BIH-Center for Regenerative Therapies, Berlin Institute of Health, Charité-Universitaetsmedizin Berlin, 13353 Berlin, Germany; 3Center for Musculoskeletal Surgery, Charité-Universitaetsmedizin Berlin, 13353 Berlin, Germany; kathi.thiele@charite.de (K.T.); markus.scheibel@kws.ch (M.S.); 4Department Shoulder and Elbow Surgery, Schulthess Klinik, 8008 Zurich, Switzerland; 5Experimental Trauma Surgery, Department of Trauma, Hand and Reconstructive Surgery, Jena University Hospital, Friedrich Schiller University Jena, 07747 Jena, Germany

**Keywords:** subacromial bursa, bursitis, progenitor cells, inflammation, pain, augmentation

## Abstract

The subacromial bursa has long been demolded as friction-reducing tissue, which is often linked to shoulder pain and, therefore, partially removed during shoulder surgery. Currently, the discovery of the stem cell potential of resident bursa-derived cells shed a new light on the subacromial bursa. In the meanwhile, this neglected tissue is gaining more attention as to how it can augment the regenerative properties of adjacent tissues such as rotator cuff tendons. Specifically, the tight fibrovascular network, a high growth factor content, and the large progenitor potential of bursa-derived cells could complement the deficits that a nearby rotator cuff injury might experience due to the fact of its low endogenous regeneration potential. This review deals with the question of whether bursal inflammation is only a pain generator or could also be an initiator of healing. Furthermore, several experimental models highlight potential therapeutic targets to overcome bursal inflammation and, thus, pain. More evidence is needed to fully elucidate a direct interplay between subacromial bursa and rotator cuff tendons. Increasing attention to tendon repair will help to guide future research and answer open questions such that novel treatment strategies could harvest the subacromial bursa’s potential to support healing of nearby rotator cuff injuries.

## 1. Introduction

Bursae are sac-like synovial structures that function as friction-reducing cushions at locations with high mechanical loading experience such as joints [1]. The subacromial respective subdeltoid bursa is the largest bursa in the human body, located at the shoulder laying between the acromion, deltoid, and the rotator cuff tendons (Figure 1). With such anatomical positioning, the bursa is in a clinically relevant location and linked to increasing numbers of musculoskeletal surgeries. Shoulder pain is among the most frequent musculoskeletal complaints, specifically in an aging society but also affects younger individuals such as athletes. Specifically, the subacromial bursitis, an inflammation of the subacromial bursa, is a common cause for shoulder pain [2,3,4]. There are manifold reasons for the development of a subacromial bursitis including recurrent overuse, acute trauma, subacromial impingement, calcific deposits, infection, autoimmune diseases, and inappropriate vaccination agent application (SIRVA, “shoulder injury related to vaccine administration” [5]). The latter being particularly relevant also in the current COVID-19 pandemic [6,7,8,9]. Treatment of bursitis is often performed conservatively using physiotherapy or injection of anti-inflammatory drugs in mild and moderate cases. The removal of parts of the bursa (partial bursectomy) is also a common procedure in severe cases that do not respond to conservative treatments [10]. In the majority of cases, a subacromial decompression is performed as well. If bursitis reaches a chronic stage, it can harm the surrounding structures, such as the supraspinatus (SSP) tendon of the rotator cuff, leading to tendinopathy or even tendon tear. However, it still must be elucidated if subacromial bursitis is the cause or consequence of an SSP tear, as both pathologies often occur simultaneously. An example of the appearance of bursitis in an arthroscopic image compared to a healthy bursa is depicted in Figure 2. Next to its role in the pathogenesis of shoulder pain, recent studies claim the bursa as a healing-promoting structure due to the presence of high proportions of stem cells, vascularization, and growth factors [11,12,13,14].

There still exists a controversy on the role of the subacromial/subdeltoid bursa, whether it leads to the development of shoulder pathologies or is supporting its regeneration [15,16]. In consequence, different surgical strategies are chosen that either lead to no or partial resection of the subacromial bursa, depending on whether the bursa is perceived as a friend or foe of the shoulder pathology. Pre-clinical studies on bursa tissues and its cells arise and provide increasing evidence that the bursa could be an active contributor to improving, specifically, rotator cuff tears in the future [17,18].

## 2. History of Attention towards the Subacromial Bursa

Already in 1872, the clinical condition of sudden pain and the stiffness of the shoulder was described by Duplay and called “periarthritis humeroscapularis”. In 1934, EA Codman published the book “The Shoulder”, which laid the foundation for a significant understanding of the subacromial bursa and the pathogenesis of bursitis [19], the painful inflammation of the subacromial bursa that leads to swelling and restricted movement of the shoulder joint. Codman was the first to shed light on the role of rotator cuff tendons in the pathogenesis of subacromial bursitis. He described a clear relationships between rotator cuff tears and pathological changes in the subacromial bursa such as calcified particles, villus formation, inflammation, fluid, bursal adhesion, and strap formation [19]. Most of the publications at that time described the subacromial bursa in a clinical context with pain and restricted movement. The studies report on clinical and radiographic observations of painful shoulders, describing the possible pathogenesis of subacromial/subdeltoid bursitis with the appearance of calcified deposits in the tendon and bursa and invasion of immune cells [19,20,21,22,23]. Just recently, these calcific deposits were characterized and identified as being nanometer-sized rod-like crystals, mainly composed of hydroxyapatite. A fragmentation of these crystals as fine needle-like structures has a high potential to induce severe inflammation [24]. It was early recognized that the “floor” of the subacromial bursa, meaning the region that is closely located to the tendons, is the center of pathological changes [19,21]. The choice of treatment was primarily dependent on the severity of symptoms and the presence of calcified deposits in the bursa and the rotator cuff tendons [23]. The surgical excision of the calcified deposits was described as the treatment of choice in acute conditions resulting in immediate and permanent pain relief [21,23]. Other proposed treatment options for acute to chronic shoulder pain resulting from subacromial bursitis in the 1930–1950s included multiple needling and puncture of the bursal fluid [20,25] and injection of local anesthetics such as Novocain, or Procaine [20,26]. Non-invasive therapies used included heat and physical therapies [20], X-ray therapy with medium to high voltage X-rays [20,26,27], and ultrasonic waves [28,29]. The rationale of X-ray therapy and needling was explained by an increase in vascularization, which was hypothesized to reduce the tension and, thus, stiffness of the tissue [20]. Altogether, next to surgical excision, X-ray therapy was described to have the best efficacy at that time [20,30]. Despite the promising results, X-ray therapy disappeared in the 1960s due to the discovery of anti-inflammatory drugs and knowledge about X-ray-induced diseases [31]. From the 1950s on, increasing studies arose on direct treatments of the bursal inflammation with injection of cortisone analogues such as corticotropin [30] or hydrocortisone acetate [29,32]. In most cases, these anti-inflammatory treatments resulted in complete pain relief and return to normal function. In 1972, Neer introduced the term “impingement syndrome”, a painful pinch of soft tissues under the acromion during movement. He proposed a new treatment strategy, anterior acromioplasty, which reduces the subacromial compression by enlargement of the subacromial space [33]. Therefore, in severe cases that failed in healing or providing pain relief after conservative treatment, a subacromial decompression including partial bursectomy was a common scenario [34]. What remains from all the mentioned treatment options today are ultrasound and physiotherapeutic treatment and corticosteroid injections as well as partial bursectomy. In the case of rotator cuff tendinopathy, subacromial injections of platelet-rich plasma (PRP) may also be justified by a possible beneficial effect of the biological components of PRP. PRP’s involvement in pain reduction is likely, but details of its benefits are still under debate [35,36,37,38]. Furthermore, the effectiveness of hyaluronic acid injections, with proposed anti-inflammatory effects [39], were investigated in placebo controlled clinical trials, but could not compete with the effects of corticosteroids in the treatment of subacromial bursitis [40,41]. Nutritional supplementation is gaining popularity as a support in the treatment of tendinopathies [42,43,44], but specific results on its role in subacromial bursitis do not exist so far. Altogether, there is still inconclusive evidence of the efficacy regarding these different conservative and invasive treatment options as summarized in a systemic review [45]. Regarding the value of subacromial decompression, there still exists a controversial debate. Some clinical cohort studies support the value of subacromial decompression for improving clinical outcomes compared to pre-surgical situations and physical therapy alone [46,47]; whereas in a multicenter, randomized, placebo-controlled trial, subacromial decompression resulted in no benefit for the patient regarding pain and function 6 months and 1 year after surgery compared to placebo surgery [48]. Furthermore, in a randomized controlled trial, the long-term outcome for more than 10 years was compared in patients with rotator cuff tendinopathy receiving supervised exercise treatment alone or in combination with subacromial decompression, and no differences were found [49]. Finally, a recent systemic review concluded that subacromial decompression does not beneficially improve clinical outcomes for patients with subacromial pain compared to placebo surgery and physiotherapy only [50]. In Figure 3, the publication records of all PubMed-listed publications around the subacromial bursa from 1934 to 2021 are summarized.

## 3. Cells Come into Focus

In 1983, the composition of resting cells in the subacromial bursa was mentioned by Sarkar and Uhthoff describing an increase in cells in the bursal wall as well as proliferation of endothelial cells in the vasculature, indicating a proliferating degenerative phenotype of the subacromial bursa in the case of rotator cuff pathology. However, inflammatory leukocyte cell infiltrates were not identified as a common phenomenon [51]; whereas later on, others reported on the presence of T-lymphocytes and monocytes/macrophages in patients with impingement syndrome and pain at rest but not in patients experiencing pain only during movement [52]. Currently, it is not debatable that immune cells are a relevant cell type in subacromial bursa tissue, which is not only the case in pathological conditions but also in healthy bursae to a lesser extent [53]. This seems to indicate that the bursa can serve as a reservoir for relevant immune cell populations. However, to date, it is totally unclear how these cells and their release of inflammatory mediators are affecting adjacent tissue such as rotator cuff tendons.

In 2013, the mesenchymal stem cell (MSC) properties of bursa-derived cells were discovered, indicated by the expression of stromal-cell-associated markers and a multipotent differentiation potential into the adipogenic, osteogenic, and chondrogenic direction [54,55]. Later, others also confirmed the presence of resident MSCs in the human subacromial bursa tissue, which are similar or even better compared to bone marrow MSCs [17,56,57]. Next to the differentiation potential into the adipogenic, osteogenic, and chondrogenic direction, the cells also exhibited a tenogenic differentiation with increased expression of the tendon-associated markers scleraxis, tenomodulin, and tenascin C [18]. This tenogenic differentiation potential makes the subacromial bursa tissue even more interesting for its potential to support healing of the adjacent rotator cuff tendons. Bursa-derived stem cells also exhibited a neurogenic differentiation potential, identifying these cells as a more potent alternative source compared to bone marrow MSCs [58]. The isolation method either using enzymatic digestion or mechanical chopping, in all cases, provided sufficient MSC populations [59], but the age of the donors significantly affected the time to form colonies in culture [60], which is a typical phenomenon also for tenocytes of the rotator cuff [61,62]. Others reported that it was not patient characteristics but the size of the rotator cuff tear that affects the colony forming unit potential [63]. Altogether, the stem cell potential seems to be the most striking reason to preserve the subacromial bursa during shoulder surgery, as it can function as a reservoir for healing-promoting cells. Recently, we were able to show a new property of bursa-derived cells: a mechano-responsiveness. Mechano-transduction pathways, ECM formation, and matrix remodeling processes were activated as a reaction to physiological and/or pathological loading conditions. These hint at a physiological function of mechanical loading in bursa tissue, which should be considered in future studies investigating the role of the bursa in the shoulder joint [57].

## 4. Bursitis and Pain: A Clear Correlation

The contribution of inflamed bursa tissue in shoulder pain is well accepted. This is supported by the fact that the subacromial bursa is innervated by several nerves, such as the suprascapular nerve and the lateral pectoral nerves and contains free nerve endings [64,65,66]. Thereby, neural structures are absent in the intima and found primarily in the subintima of the bursa in close proximity to blood vessels [67]. The presence of the neuropeptide substance P was rated as the reason for shoulder pain in patients with rotator cuff disease [68]. In addition, the expression of vascular endothelial growth factor (VEGF) in the subacromial bursa was found to be associated with motion pain in patients with rotator cuff tear or subacromial bursitis [69]. Additionally, it was demonstrated that inflammatory conditions in the synovium of the glenohumeral joint do not contribute to pain in rotator cuff disease but must arise from the bursal side [70]. From a histopathological point of view, it was described in rotator cuff tear patients that bursal inflammation, necrosis, hypertrophy, and oedema correlated with pain, contributing to the role of the subacromial bursa as a pain generator [12]. In summary, the presence of nerves and pain mediators in the subacromial bursa provokes most authors to conclude that a partial bursectomy helps to eliminate pain from the shoulder.

## 5. Bursa as a Friend or Foe for Rotator Cuff Repair

The properties of the subacromial bursa and its scientific interpretation let the subacromial bursa appear either as a friend or a foe for the development of tendon pathologies or its regeneration. The bursa was claimed as a possible reason for tearing due to the presence of active bone morphogenetic proteins (BMPs), such as BMP-2 and BMP-7, which might induce ectopic bone/cartilage formation and promoted tendon tearing. Therefore, the authors postulated the early removal of the subacromial bursa for better healing outcomes [71]. However, recently, no differences in BMP-2 expression were found in bursa tissue of healthy donors compared to different shoulder pathologies including partial and full-thickness SSP tear [53]. The presence of pro-inflammatory cytokines and cells in the bursa tissue often lays the basis for a negative association of the subacromial bursa in rotator cuff repair. It has been described that inflammatory cytokines, such as SDF-1, IL-1β, TNF-α, TGF-β, and IL-6 as well as pain-associated mediators (cyclooxygenase 1 (COX-1) and 2) and proteolytic enzymes (matrix metalloproteases (MMPs)) are increased in subacromial bursae from patients with rotator cuff tear or frozen shoulder compared to healthy controls [14,53,72,73,74,75,76,77,78]. Most authors postulated from this result that a partial bursectomy is necessary during rotator cuff surgery to reduce pain and inflammation and, thus, improve rotator cuff repair. However, only COX-2 expression in the subacromial bursa at the time of surgery was associated with failed rotator cuff healing 6 months post-surgery, whereas several other markers of inflammation, angiogenesis, and remodeling were not linked to healing [79]. Benson et al. investigated the cellular composition of the subacromial bursa of patients with impingement syndrome receiving a subacromial decompression. They found no correlation of histomorphometry regarding cell proliferation, fibroblast cellularity, vascularity, or the appearance of macrophages, leukocytes, mast cells, amyloid, or chondral metaplasia with the outcome after subacromial decompression [80]. Concluding from that, the pure quantity of inflammatory cells and bursa fibroblasts in the subacromial bursa does not influence the healing outcome.

The subacromial bursa is a well vascularized structure [13] with a vessel density of 1.8–3.4%, depending on whether the floor or roof of the bursa is being investigated [67]. Due to the location of the subacromial bursa, its tight fibrovascular network with high proportions of proliferating cells is also covering the tendon in the case of a tendon tear. Uhthoff and Sarkar found more signs of tissue repair rather than markable degeneration or necrosis of the bursa. Therefore, the preservation of the subacromial bursa as an important healing-promoting structure in the shoulder joint was advocated due to the fact of its fibrovascular properties [1]. Similar observations were made by others, supporting the localized bursal reaction at the tendon rupture side as a necessary feature for an optimal tendon repair after surgical reconstruction [11]. Moreover, the presence of fibrosis, the thickening of connective tissue in the bursa, was associated with a better clinical outcome after anterior acromioplasty for impingement syndrome [81]. In contrast to the assumption that inflammation is the driver of rotator cuff pathology, others interpreted the increase in bursal inflammation with the attempt of the bursa to activate rotator cuff repair. More interestingly, a decrease in tendon neo-angiogenesis and inflammation was found with age, but vice versa an increase in neo-angiogenesis, inflammation, and hypertrophy in the bursa was observed. Therefore, it was speculated that the bursa is taking over the reparative processes in the elderly [82]. A further benefit of the bursa for rotator cuff healing might be the presence of resident cells with progenitor potential in the bursa. This has led several authors to conclude that the bursa might be a promising tool in augmentation of rotator cuff repair [17,55].

## 6. Animal Models for Understanding the Role of the Bursa

For pre-clinical assessment and evaluation of therapeutic candidates, animal models are of great importance. They allow us to study the detailed role of the subacromial bursa inflammation and its impact on tendon healing. Recently, the subacromial bursa in the rat was characterized to have similar properties compared to the human bursa having a more fibrous structure beneath the acromion and is fattier in its medial portion [83]. Furthermore, the MSC potential of rat and mice bursa-derived cells is comparable to human bursa-derived cells [83,84], which provides the clinical relevance of such models. The interplay between tendon and bursa was also demonstrated in this model by showing that a bursectomy reduces the inflammatory response in the injured tendon [83]. A study in mice that received an SSP detachment without repair showed the importance of the cells on the bursal side of the SSP tendon in rotator cuff healing. However, the origin of these cells could not be fully identified, whether they come from the paratenon, bursa, or overlying muscle [85]. In rabbit and chicken rotator cuff repair models, the subacromial bursa significantly contributed to the tendon healing process, indicated by high cell proliferation at the bursal side of the tendon [86,87].

Experimental animal models represent the basis for testing strategies to target or better understand the bursa tendon interaction. MicroRNA-29a seems to play a role in the pathogenesis of bursal inflammation and fibrosis in patients with rotator cuff tear with shoulder stiffness. The microRNA-29 family members were described to inhibit the expression of fibrotic markers and extracellular matrix production (He et al., 2013). Therefore, the experimental approach of Ko et al. provided promising evidence of improved rotator cuff integrity and shoulder function in transgenic mice overexpressing microRNA-29a by repressing fibrosis of the subacromial bursa through direct targeting of Col3A1 mRNA [88]. Next to in vitro multipotent differentiation, in vivo implantation in mice of bursa-derived MSCs that were subjected to BMP-12 and seeded onto ceramic scaffolds also formed extensive bone and a tendon-like tissue, which underlines the potential of these cells for tendon repair [18]. Furthermore, in a patella tendon defect model in mice, bursa-derived MSCs that were incorporated into fibrin gels showed superior engraftment into the host tissue and a greater cell survival compared to bone marrow MSCs [89]. Bursa-derived MSCs might therefore be a preferable solution for a cell-assisted tendon repair.

## 7. Possible Therapeutic Targets

With the knowledge of increased inflammatory mediators in diseased bursa tissue, experimental studies have focused on strategies to target and understand bursal inflammation. The expression of SDF1, a chemokine that stimulates the recruitment of inflammatory cells, decreased in bursa cells by COX-2 inhibitors and steroids, providing a possible therapeutic strategy to treat subacromial bursitis [74]. Furthermore, isolated bursa cells respond to IL-1β stimulation with increased expression of SDF1, which indicates a possible mechanism of these mediators in subacromial bursitis [73]. Using epigallocatechin-3-gallate, a polyphenol in green tea, the IL-1β-induced expression of pain-associated factors and pro-inflammatory cytokines could be reversed in bursa-derived cells, leading to a reduced pro-inflammatory response and expression of pain mediators in tenocytes. This might provide a new strategy to treat pain in rotator cuff tendinopathy [78]. Another option to treat the IL-1β-induced pain mediation in the subacromial bursa could be to target the transforming growth factor (TGF)-β-activated kinase 1 (TAK-1). The TAK-1 inhibitor (5Z)-7-oxozeaenol decreased the expression of the important pain-associated mediators COX-2 and nerve growth factor (NGS) in bursa-derived cells [90]. Furthermore, TNF-α could be a potential target. Here, a lentivirus-mediated RNA interference of TNF-α in a rat rotator cuff disease model provoked a decreased production of the inflammatory mediators NF-κB, MMP-1, MMP-9, COX-1, COX-2, and SDF-1, suggesting this as a potential option to decrease inflammatory-dependent pain in the subacromial bursa [91]. To improve cell migration potential and proliferation, subacromial bursa-derived cells were stimulated with a one-time dosage of insulin or insulin-like growth factor (IGF-1), which might be useful for achieving an improved healing through the provision of a sufficient number of healing-promoting cells at the rotator cuff repair side. However, no significant impact of insulin or IGF-1 on cell proliferation and migration of bursa-derived cells could be detected [92].

## 8. Clinical Strategies to Use the Augmentation Potential of the Subacromial Bursa

According to the mentioned promising properties of the subacromial bursa that could be beneficial for the augmentation of rotator cuff repair, some authors already postulated arthroscopic surgical strategies. Freislederer at al. described an easy and reliable surgical technique for bursal augmentation during repair of rotator cuff tears using an arthroscopic double-row suture technique with suturing of the lateral bursa to the tendon footprint. We postulate that the highly vascularized subacromial bursa could serve as a stimulating factor also to re-vascularize the re-attached and strangulated rotator cuff tendons [93]. A similar technique was used by Bhatia, who also prepared a bursa–tendon unit but with suturing of the posterior and posterolateral part of the bursa that covered a larger tendon area, possibly resulting in a more resilient repair [94]. A different approach describes the harvesting of bursa tissue and re-implantation of minced bursa tissue to the repair side. This is a further feasible method for bursa augmentation in rotator cuff repair, which can be used in combination with various surgical repair techniques [95,96]. In a preliminary study, the outcome after a more complex biological augmentation of rotator cuff repair using a stable clotted mixture of concentrated bone marrow, PRP, platelet-poor plasma (PPP), and subacromial bursa tissue was investigated. Using this technique, the patients’ functional outcome after rotator cuff repair significantly increased compared to pre-surgery after a minimum of 1 year follow up with a 93.8% substantial clinical benefit [97]. With the lack of a control group and the mixture of various potentially repair-promoting factors in the clot, it cannot be concluded what impact the subacromial bursa has on this approach. All authors hypothesized that the highly vascularized bursa tissue with significant proportions of repair-promoting stem cells has a strong potential in enhancing rotator cuff healing. In general, more clinical evidence is needed to evaluate the success of these promising techniques for the healing outcome of the patients.

## 9. Future Perspectives

The contribution of immune cells and pro-inflammatory cytokines in bursitis and pain is undeniable. However, it can be speculated that the bursa can serve as reservoir for important inflammatory cells and mediators that are able to initiate and promote tendon tissue repair. Therefore, a major task would be to find strategies to compensate and use inflammatory conditions in the bursa to initiate or promote healing of the nearby rotator cuff tendons. There remains a conflict between pain generation and healing promotion, and it should be carefully considered if there are other options to eliminate pain than removal of the bursa tissue. Some experimental strategies provided promising ideas to dampen pro-inflammatory and pain-generating processes, such as targeting the IL-1β or TNF-α pathways [78,90,91], which might help to overcome pain without a bursectomy in the future. However, the use of anti-inflammatory drugs should be handled with caution, as next to the wanted reduction in pro-inflammatory factors, inflammation-resolving mediators are also suppressed [98]. This is important to consider as processes such as the resolution of inflammation, which is actively mediated by specialized pro-resolving mediators [99], could be an interesting target and has not been studied in the subacromial bursa so far. The contribution of pro-resolving mediators, such as lipoxin, annexins, or resolvins, in the progress of rotator cuff tendinopathy was studied previously [100] and might also be a promising approach to dampen inflammation in bursa tissue in the future. Furthermore, what remains inconclusive from the literature is the direct interplay between the subacromial bursa and the rotator cuff tendons with the core question: is bursal inflammation the cause or consequence of rotator cuff pathologies? In addition, the role of mechanical forces in disease mechanisms remains unclear, even though the bursa serves as the important friction-reducing structure at the shoulder joint. We therefore speculate that mechanical forces are important triggers of inflammatory processes in the bursa and should be investigated in more detail in the future. The enrichment of stem cells, vasculature and growth factors at the tendon rupture site through bursal augmentation seems to be promising and easy and reliable surgical techniques for this augmentation were recently described [93,94,95,96]. However, functional follow-up results of clinical cohort studies comparing conventional rotator cuff repair to bursa-augmented strategies are needed to judge its effectivity.

## 10. Summary

In summary, biological evidence of the role of the subacromial bursa has been increasing over the last decades, and the existing knowledge as well as open questions are summarized in Figure 4. Starting as a predominant pain generator, the subacromial bursa has increasingly been considered over the last years as a promising healing-promoting structure in the shoulder joint, particularly because of the strong stem cell potential of the bursa-derived cells and distinct vasculature and growth factor content. A consensus on the role of inflammation has not yet been found, and future strategies should focus on eliminating pain but also on perceiving inflammatory mediators or resolution of inflammation pathways that might initiate or promote tendon healing. In general, further experimental and clinical research is still needed to fully elucidate the contribution of the subacromial bursa in the development and progression of tendon pathologies or its healing-promoting properties in the shoulder joint.

## Figures and Tables

**Figure 1 cells-11-00663-f001:**
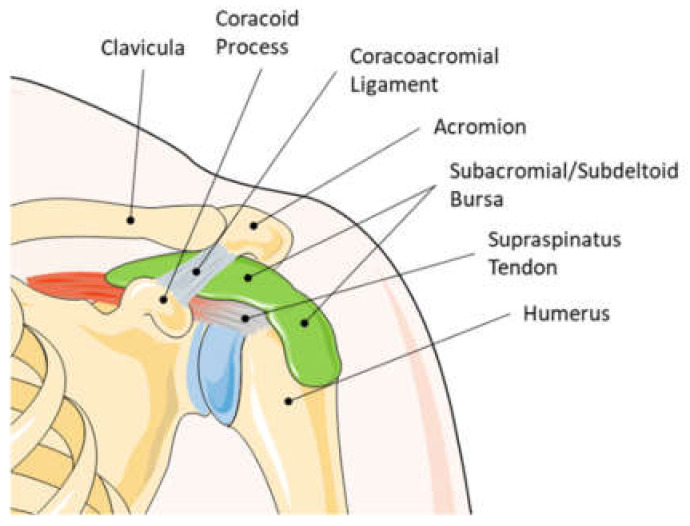
Anatomy of a left human shoulder joint. The subacromial/subdeltoid bursa lays between the rotator cuff, the deltoid, and the acromion. Due to the close location to the rotator cuff tendons, the contribution of the subacromial bursa in the development or healing of tendon pathologies seems obvious. This graphic was created with an image from Servier Medical Art.

**Figure 2 cells-11-00663-f002:**
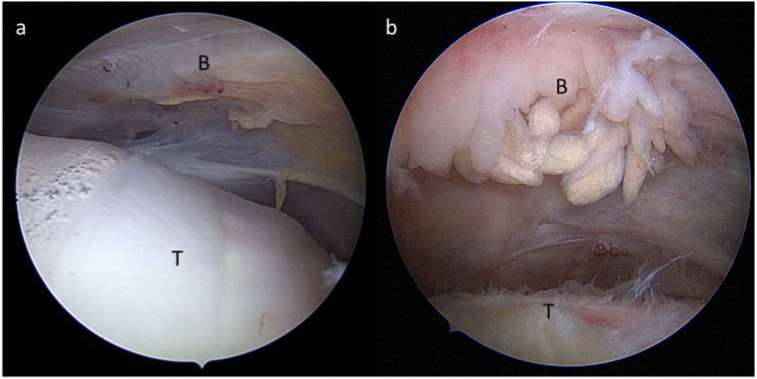
Arthroscopic view from a posterior portal during arthroscopy of a right shoulder showing (**a**) a normal subacromial/subdeltoid bursa (B) and healthy tendon (T) and (**b**) a severe hypertrophic subacromial bursitis (B) and fraying of the supraspinatus tendon (T).

**Figure 3 cells-11-00663-f003:**
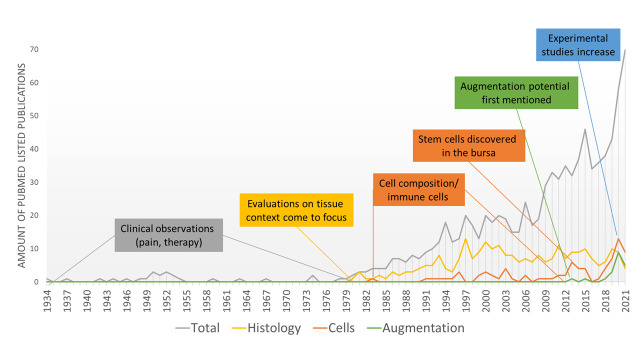
Overview of historical steps towards increased experimental research around the subacromial bursa: number of PubMed listed publications from 1934 to 2021 mentioning subacromial bursa OR subacromial bursitis (total) combined with the search terms “Histology”, “Cells”, and “Augmentation”.

**Figure 4 cells-11-00663-f004:**
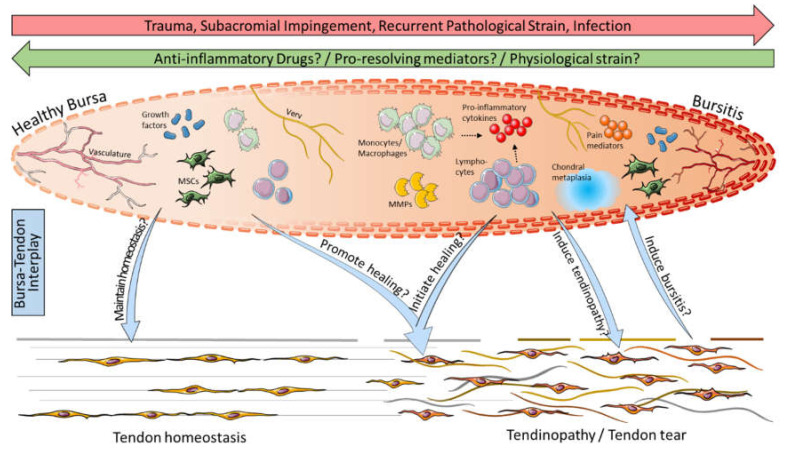
Possible bursa–tendon interaction: Inflammation through trauma, impingement, recurrent pathological strain, or infection leads to the development of bursitis, whereas bursitis might be cured by anti-inflammatory drugs, pro-resolving mediators, and/or physiological strain. Bursitis is characterized in particular by the presence of immune cells such as monocytes/macrophages and lymphocytes (NK cells, B cells, and T cells), pro-inflammatory cytokines (e.g., SDF-1, IL-1β, and TNF-α), matrix metalloproteinases (MMPs), pain mediators (e.g., COX-2, NGF, and substance P), and chondral metaplasia. Compared to that, healthy bursae contain lower numbers of immune cells and predominate by growth factors and mesenchymal stem cells (MSCs) and, thus, might maintain tendon homeostasis. In bursa–tendon interplay it is hypothesized that growth factors and MSCs in the bursa can promote the healing of tendinopathies, including rotator cuff tears, and it has to be elucidated if pro-inflammatory cells and mediators might help to initiate healing cascades at the tendon rupture side. Vice versa, it is still debatable if bursitis is the cause or consequence of pathological conditions in the tendon. This graphic was created with images from Servier Medical Art.

## Data Availability

Not applicable.

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
