# Peer review of "Subacromial Bursa: A Neglected Tissue Is Gaining More and More Attention in Clinical and Experimental Research"

_cells, 2022, doi:10.3390/cells11040663_

Round 1
Reviewer 1 Report
Dear Authors,
I found this paper interesting and well prepared. Unfortunately, as a review, it does not provide any novel data. I would recommend reedition of the last paragraph. Extraction of the summary beyond "perspectives" would clarify the structure of the manuscript. It is Editor's decision if it suites Cells requirements as a review, for such high IF journal.
Reviewer 2 Report
This article is interesting and focus on a frequent clinical topic, as a growing number of patients present in clinic with shoulder overuse problems. The Authors underline the controversial role of subacromial bursa in overuse sindrome/shoulder-tendon-chronic-lesions and analyzed the possible positive role of bursal MSCs in the repair process. Moreover They underline the role of surgery (open or arthroscopic one) and conservative therapies (i.e. physiotherapy, steroids administration, PRP, cox-2 inhibitors and other NSAID) to treat bursa/tendinopathy related pain.
On the other hand, in the clinical practice the use of topic intra-bursal hyaluronic acid supplementation are gaining popularity as treatment of chronic bursitis or small partial rotator cuff tendons lesions. Usually oral administration of specific food supplements are also recommended, specially in elderly, with variable results.
The English in fluent and the reading enjoyable.
I suggest to add, if possible, clinical or sub-clinical studies about the role of hyaluronic acid and specific food supplements as treatment of bursa/tendinopathy diseases and clarify if there are scientific evidences that support their use.
